Toxicity of TiO2, SiO2, ZnO, CuO, Au and Ag engineered nanoparticles on hatching and early nauplii of Artemia sp.

Rekulapally Rohit
Murthy Chavali Lakshmi Narsimha
Idris Mohammed M. idris@ccmb.res.in
http://orcid.org/0000-0001-5117-704X Singh Shashi shashis@ccmb.res.in
Centre for Cellular and Molecular Biology , Hyderabad, Telangana , India
Mortimer Monika
Electronic publication date: 2019 Jan 3
Publication date: 2019
Volume: 6
Electronic Location ID: e6138
Received 2018 Jul 18; Accepted 2018 Nov 20
Copyright: © 2019 Rekulapally et al.
Copyright year: 2019
Copyright holder: Rekulapally et al.
License: This is an open access article distributed under the terms of the Creative Commons Attribution License, which permits unrestricted use, distribution, reproduction and adaptation in any medium and for any purpose provided that it is properly attributed. For attribution, the original author(s), title, publication source (PeerJ) and either DOI or URL of the article must be cited.
License URL: https://creativecommons.org/licenses/by/4.0/

Keywords: Nanomaterial, Toxicity, Hatching, Method, Nauplii, Salt water

Funding: European Union FP7 263147 NANOVALID This work was supported by European Union FP7 project no. 263147 NANOVALID. There was no additional external funding received for this study. The funders had no role in study design, data collection and analysis, decision to publish, or preparation of the manuscript.

==============================
The potential of environmental release enhances with increased commercial applications of the nanomaterials. In this work, a simple and efficient test to estimate the acute toxicity of nanoparticles is carried out on Artemia species and their hatching rates. We have tested six different engineered nanoparticles (silver, gold, copper oxide, zinc oxide, TiO2 and SiO2 nanoparticles) and three soluble salts (CuSO4, ZnSO4 and AgNO3) on Artemia sp. The physicochemical properties of the nanoparticles involved in this study were analyzed in normal water and marine water. Hydrated and bleached Artemia cysts were allowed to hatch in continuously aerated, filtered sterile salt water containing nanoparticles; hatching of viable nauplii and total hatchlings have been recorded. In parallel, standard Artemia toxicity test was conducted on the nauplii monitoring the viability. In hatching experiments, a reduction in hatching rate was observed along with mortality of newly hatched nauplii. The results of the hatching experiment and of the standard Artemia test showed a good correlation. The toxicity of the nanoparticles was compared and the order of toxicity was estimated as Ag>CuO>ZnO>Au>TiO2>SiO2. The study thus suggests that the hatching test itself is a reliable assay for determining the toxicity of nanomaterials.

Introduction

Nanomaterials with their ever-expanding diversity, unique properties and endless applications pose risk to environment and human health. There is a dearth of information on the impact or the risk of nanoscale objects to the environment. Increasing utility would mean enhanced exposure of the ecosystems to these unknown risks. Their release into the environment may start with production, during their applications, by weathering and finally through wastes.

Nanomaterials can be designed from almost any material—metal/oxides, carbon, organic, biomaterial etc., and in any form, by strictly adhering to the size range of nanometer at least in one dimension. Nanomaterials behave differently from the bulk due to their unique physicochemical properties. Information on predictability of their behavior or state in due course is almost negligible. Not only is there limited information on the bioavailability and biopersistence of these particles, the behavior of the particles in various conditions like pH, salinity and other biotic factors is not clearly known (Musee, 2011; Moore, 2006). Efforts are currently on to identify the risks associated with the nanomaterial applications.

Hazard identification is an important step for risk assessment of nanomaterials. Most of the methodologies used in hazard identification are the ones used for chemicals in general. Being different from bulk material, the nanoforms of the same materials may pose some restraints or advantages in the use of methodologies.

Aquatic systems are the inevitable receptacles for the materials released in the environment. The organisms therein get exposed to the nanomaterials accumulated from seepage and flow through. Many assays for ecotoxicology have been tested for nanomaterials—using bacteria (Suppi et al., 2015), fishes and fish embryos (Rizzo et al., 2013; Zhu et al., 2008); copepods (Templeton et al., 2006); Daphnia (Adam et al., 2014; Völker et al., 2013). In this study, Artemia sp. are used to study the nanomaterial toxicity. Artemia is a nonselective filter feeder organism which is available worldwide in highly saline waters. Due to their ubiquitous distribution, robust nature and cost-effective culturing conditions they make a good model organism for toxicological assays. Though Artemia are considered to be one of the insensitive models for a lot of chemicals, the early developmental stages of Artemia are highly vulnerable to many of the test materials. Few studies are reported for nanomaterial toxicity using Artemia sp. also (Arulvasu et al., 2014; Ates et al., 2012, 2013, 2015; Rajabi et al., 2015). Selection of end point is a factor worth considering while evaluating sensitivity of the species. Most of the studies are carried out in newly hatched nauplii that are exposed to varying concentration of nanomaterial; hatching of hydrated cysts has not been used as an end point for toxicity except in (Carballo et al., 2002; Migliore et al., 1997; Rotini et al., 2015; Sarabia et al., 2008) though some earlier studies had shown sensitivity of hatching rate of Artemia to metals (Go, Pandey & MacRae, 1990; Sarabia et al., 1998; Brix et al., 2006).

In this study, the nanomaterial we have evaluated are metals and metal oxide nanoparticles (NP); these have a high production globally due to their demand for diverse applicability. Generally, these nanomaterials find widespread applications as their corresponding bulk materials are nontoxic.

Acute toxicity of four metal oxide nanoparticles (TiO2, SiO2, ZnO and CuO) and two metal nanoparticles (Ag and Au) to nauplii of Artemia and on the hatching of their hydrated cysts has been studied. For the standard Artemia test and hatching test, mortality and hatching rates were considered as end points. Both the tests exhibit a good inverse correlation in sensitivity to nanoparticles, thereby implying that a bioassay based on hatching can be developed for hazard assessment of nanoparticles.

Materials and Methods

Preparation of nanoparticles and their characterization

All the nanoparticles used in the study were obtained from the consortium of the NANOVALID (Nordmiljö AB, Sundsvall, Sweden) project. SiO2 nanoparticles (powder form) were supplied by Nanologica AB (Södertälje, Sweden), silver nanoparticles were obtained as a suspension of 4% from Colorobbia, Firenze, Italy; gold nanoparticles as a suspension of 0.006% from INMETRO (Rio de Janeiro, Brazil); copper oxide nanoparticles (powder form) from Intrinsiq Materials (Hampshire, UK); zinc oxide nanoparticles (powder form) from Nanogate (Quierschied-Göttelborn, Germany) and TiO2 particles were synthesized in house using the existing protocol (Reddy et al., 2004). The nanoparticles obtained in powder form were made into a suspension. A suspension of NPs was made by suspending SiO2, TiO2, CuO and ZnO NPs at a concentration of two mg/mL. The suspension was sonicated at ∼30 W with pulse of 50% for 10 min. Silver nanoparticles were obtained as suspension at a concentration of 4% with polyvinylpyrrolidone (PVP) as a stabilizing agent, gold nanoparticles in water at a concentration of 0.006%. Working dilutions of the nanoparticles were prepared in salt water (SW) (Oceanfish marine salt; Prodac International, Cittadella PD, Italy) for the experiments.

The nanoparticles used in this study were characterized in the NANOVALID consortium in round robin manner. Transmission electron microscopy (TEM) and dynamic light scattering (DLS) were used in this study for basic characterization of nanoparticles before every experiment. In TEM studies the nanoparticle suspension of 50 μg/mL was applied to formvar coated copper grids and air dried. The grids were examined in JEOL 2010 TEM at 100 kV using 20 μ aperture, about 10 images were taken using GATAN camera and later analyzed using GATAN software. The images of about 100 particles were used for analysis. The nanoparticle suspensions at concentration of 25 μg/mL in milliQ water (mQW) and SW were measured in DLS system (Horiba Nanopartica SZ100, Kyoto, Japan). Each suspension was analyzed thrice up to 48 h and the data was represented as mean ± SEM. Solvents used to prepare the nanoparticles were used as controls for size determination.

Hatching of Artemia

For every one L culture, one g of Artemia cysts were used (San Francisco Bay Strain, UT, USA). These cysts were hydrated in distilled water under aeration for 45 min and then bleached to decapsulate the cysts wall using 20% sodium hypochlorite solution for 15 min. The bleached Artemia eggs were then washed for six times with distilled water and allowed to hatch in 2.5% SW (pH 8.0) with continuous light and aeration for 24 h (Sorgeloos, 1973). The temperature was maintained at 28 °C, the optimum temperature for hatching of Artemia cysts. Nanoparticles were added to the culture in concentrations of 100 and 10 mg/L. pH of the SW containing nanoparticles was always determined using pH meter after setting the pH meters with two reference standards at pH 7 and 9.

Toxicity studies using hatching rate as end point

Six types of nanoparticles (TiO2, SiO2, ZnO, CuO, Au and Ag) are tested for their toxicity on brine shrimp hatching. The Artemia cysts were allowed to hatch under standard culture conditions and in presence of two different concentrations of each nanomaterial (100 and 10 mg/L). The hatching rate, mortality and viability were checked after 24 h for each Artemia cultures exposed to nanoparticle. Three aliquots of 100 μL was taken from each flask after thorough mixing and counted under the stereo-microscope. Both, hatched Artemia (immotile vs. motile) (h), and unhatched (u) cysts were counted. Hatching rate was expressed as percent of total number of fully hatched Artemia in comparison to total Artemia (H) in the aliquot. The experiment was repeated six times. The average and SEM was calculated for each sample.

Hatching rate=h/H×100

Standard Artemia toxicity test

In a second set of experiments, following the protocol of Solis et al. (1993), Artemia hatched under normal conditions in SW were exposed to various nanoparticles. A total of 10 nauplii were transferred to six-well plates and the nanoparticles at 100, 10, 1, 0.1 and 0.01 mg/L concentrations were added. The number of live larvae was counted at intervals of 6, 12, 18, 24, 36 and 48 h. Mortality of the larvae was used as end point. Mortality was recorded as cessation of swimming or any movement by nauplii. LC50 was calculated for each particles using Graphpad (www.graphpad.com/quickcalcs/).

Oxidative stress test

Stresses due to reactive oxidation species can be observed by treating the nauplii exposed to different concentrations with 10 μM dichloroflourescien diacetate (DCFDA). DCFDA or with 20 μM dihydroethidium (DHE), both of which are chemiluminiscent probes and react with ROS generated due to oxidative stress. The nanoparticle treated nauplii exposed to DCFDA or DHE are imaged in Zeiss Axiovert imager using appropriate filters.

Validity of results

The hatching studies have been conducted six times and the validity criterion used is, low mortality in the control. The data was analyzed for statistic validation using Graphpad (www.graphpad.com/quickcalcs/). In the Artemia toxicity tests the validity criterion was mortality below 10% in 24 h period. The correlation test was performed for both sets of data using on line software for correlation coefficient and comparing data using Microsoft excel.

Results

Analysis of nanoparticles

The size of nanoparticles both in distilled water and SW was measured. The nanoparticles showed reduced size in SW by TEM. The TEM studies showed no apparent change in aggregation or dispersion of particles (Fig. 1). Measurement of particle size by DLS which allows examination of particles sizes as well as aggregates when in suspension revealed that nanoparticles exhibited different aggregation profiles in mQW and SW. TiO2 (7,082 nm in mQW: 8,009 nm in SW), CuO (329 nm in mQW: 468 nm in SW) and ZnO (117 nm in mQW: 185 nm in SW) nanoparticles showed higher values for aggregate sizes in SW as shown in Fig. 2. Silver (169 nm in mQW: 101 nm in SW), gold (5,542 nm in DW: 3,712 nm in SW) and SiO2 (7,704 nm in DW: 6,096 nm in SW) had smaller aggregates in SW as compared to particles in mQW (Fig. 2). Visible flocculation was observed in case of copper and zinc oxide nanoparticles. DLS measurements for these nanoparticles gave very high values corroborating the flocculence seen (Fig. 2). Though size changes in the mean size of particles in mQW and SW was noticed both by TEM and DLS, it was not found to be significant in statistical analysis (Table 1).

Figure 1 TEM of nanoparticles in distilled and salt water.

Transmission electron micrograph of nanoparticles suspended in distilled water and high salt solution. Panel on the left represents particles in milliQ water and on right particles in salt water. The particles are TiO2 (A, B); SiO2 (C, D); ZnO (E, F); CuO (G, H); Gold (I, J) and Silver (K, L) nanoparticles. The average size of nanoparticles is mentioned in nm.

Figure 2 Size of nanoparticles by DLS.

Size distributions of nanoparticles in distilled water and salt water. Many nanoparticles show reduced particle size due to dispersion in salt water. Au, SiO2 (A) and Ag (B) show reduced particle size in salt water. TiO2 (A), copper oxide and zinc oxide (B) show higher values when suspended in salt water.

Table 1 Size comparison.

NP	TEM (N = 75)	p-value	DLS (N = 3)	p-value	
MilliQ	Salt water	MilliQ	Salt water	
Mean	SEM	Mean	SEM	Mean	SEM	Mean	SEM	
SiO2	332.18	98.24	238.7	41.10	0.3814	7,703	1,041	6,095	1,248	0.3786	
TiO2	20.2	6.44	10.82	0.132	0.1474	7,081	1,094	8,008	1,643	0.6628	
ZnO	14.95	2.9	12.88	11.55	0.8625	117.1	12.869	184.8	8.48	0.0118	
CuO	22.63	5.77	14.38	1.34	0.1663	328.95	58.76	467.66	72.26	0.2107	
Au	34.35	13.23	25.76	3.46	0.5310	5,542	2,457	3,712.67	341.18	0.5017	
Ag	40.21	12.19	25.33	12.21	0.3898	169.43	50.99	100.73	26.52	0.2979	
Notes:

Size comparison of the nanoparticles in milliQ water and salt water.

The nanoparticles suspended both in mQW and 2.5% NaCl solution (SW) were measured up to 48 h to observe change, if any in their dispersion pattern (Fig. S1). Silver NPs, TiO2 and SiO2 did not show much difference in dispersion in mQW over 48 h but the particles showed higher aggregate sizes in SW in 48 h. In CuO and ZnO NPs visible flocculation and sedimentation was seen in case of both MilliQ and salt solution by 48 h.

Hatching studies

Brine shrimp Artemia were allowed to hatch in filtered sterile SW kept at continuous aeration that also keeps the particles in suspension throughout the experiment. Brine shrimp showed a hatching rate of 74% ± 5.5 in mQW and the viability of these hatched nauplii was about 97%. The hatching rate of Artemia altered drastically in presence of nanoparticles (Table 2; Fig. 3). The hatching rate dropped with increase in concentration of nanoparticles in SW. The highest effect was seen in presence of silver and copper oxide nanoparticles. The hatching rate dropped to about 29% in presence of these two nanoparticles at a concentration of 100 mg/L; of these only 50% were motile and viable at a concentration of 100 mg/L (Table 2). The hatching rate remained lower at 10 mg/L of silver nanoparticles but the viability of hatched nauplii was good at 85%. With copper oxide also hatching rate was low (43.23%) with better viability around 64%. Similarly, with silver nitrate or in presence of capping agent PVP the hatching slightly improved (33% and 41%, respectively) with good viability of 70% and 84%, and about 60% in case of copper sulphate with viability of 33%.

Table 2 Hatching rate.

Hatching	Concentration of eNP	Percent hatching rate	SEM	p-value	Immediate mortality of hatchlings	SEM	p-value	
Blank		74.6	5.18		2.03	1.3		
AgNP	100 mg/L	29.3	9.978	0.013	48.31	14.51	0.010	
AgNP	10 mg/L	32.2	8.24	0.010	15.35	5.0	0.033	
AgNO3	10 mg/L	33.4	9.19	0.019	30.71	12.1	0.043	
PVP	42 μg/L	41.1	9.95	0.079	15.71	5.62	0.046	
AuNP	1 mg/L	57.1	10.41	0.622	4.31	1.83	0.46	
AuNP	0.5 mg/L	53.1	9.39	0.378	9.08	2.8	0.069	
SiO2NP	100 mg/L	48.9	5.60	0.390	20.04	11.9	0.176	
SiO2NP	10 mg/L	63.8	9.65	0.943	8.18	5.8	0.365	
TiO2NP	100 mg/L	41.51	12.68	0.147	32.5	11.4	0.028	
TiO2NP	10 mg/L	54.3	5.48	0.275	18.41	4.8	0.159	
TiO2NP	1 mg/L	59.07	6.01	0.631	2.20	0.7	0.062	
CuONP	100 mg/L	29.14	13.25	<0.0001	55.00	39.02	0.008	
CuONP	10 mg/L	43.24	17.33	<0.0001	33.82	7.41	0.054	
CuSO4	10 mg/L	59.06	11.07	0.0078	47.00	21.74	<0.0001	
ZnONP	100 mg/L	60.53	13.36	<0.0001	49.60	32.5	0.002	
ZnONP	10 mg/L	63.38	14.97	<0.0001	33.75	30.14	0.012	
ZnSO4	10 mg/L	44.72	22.27	<0.0002	42.00	37.97	0.014	
Note:

Hatching rate of Artemia cysts and immediate mortality of the nauplii.

Figure 3 Hatching rate and mortality.

Hatching rate and percent of dead nauplii in the hatching experiment. Silver nanoparticles show the highest toxicity both in terms of hatching and immediate survival of nauplii after hatching. Other nanoparticles show decline in hatching but posthatching survival is not affected. SiO2 nanoparticles at lower concentration do not affect hatching and survival. *Significance p < 0.05.

Hatching rate in presence of other nanoparticles (gold, SiO2, TiO2, copper oxide and zinc oxide) was low at about 50–60% as compared to control (Table 2; Fig. 3) and posthatching mortality was not very high. In presence of zinc oxide nanoparticles, 60% of the nauplii hatched but the cysts had a tendency to clump together. In TiO2 at 100 mg/L, hatching rate was 41.50% with 67.8% viability; the hatching rate was at 54% with ∼90% viability in 10 mg/L TiO2 dioxide NPs. Gold and SiO2 nanoparticles displayed very good viability after hatching (Fig. 3). SiO2 nanoparticles at a concentration of 10 mg/L displayed hatching rate and viability equivalent to the controls.

The pH of SW did not change upon addition of nanoparticles and remained around 8.3–8.5 up to 48 h. Live hatched nauplii appeared normal and swimming vigorously. There were no visible changes in morphology of nauplii hatched in presence of nanoparticles. Some of the nauplii had emerged but were immotile/dead; some were trapped in the membrane and still-dead (Figs. S2.1–S2.6). Silver and copper oxide nanoparticles which had maximum toxic effect on hatching rate also did not show any gross morphological abnormalities. Staining of hatched Artemia with DCFDA revealed oxidative stress in a few still trapped nauplii and the gut of dead nauplii (Fig. S3).

Larval toxicity test

Following the ARC (Artemia Reference Center) test protocol, the larvae hatched in SW were exposed to nanoparticles for up to 48 h and the viability recorded every 6 h for first 24 h and subsequently every 12 h. Artemia were not fed during the test period. The nauplii showed variability in mortality to various nanoparticles. The most toxic was the silver nanoparticles and silver in ionic form (Table 3). Silver in ionic form was highly toxic as 80% of the nauplii were dead in 24 h in silver nitrate solution (Fig. 4). The LC50 of the silver nanoparticles was around 10.12 mg/L at 24 h. Other nanoparticles used in the study were nontoxic initially but after 30 h (Table 3), Many Artemia died in concentration-dependent manner (Fig. 4). LC50 of gold was 7.54 mg/L at 48 h. LC50 of SiO2, TiO2 and CuO was 23.59; 18.94 and 20.92 mg/L at 48 h. In presence of zinc oxide nanoparticles, around 36% nauplii perished in 48 h in the highest concentration used with the LC50 above 100 mg/L at around 259.34 mg/L. Ionic forms of silver, copper and zinc were found to be more toxic than their corresponding nanoforms having LC50 Ag+ (8.19 mg/L), Cu++ (13.5 mg/L) and Zn++ (0.769 mg/L).

Table 3 Lethal concentration of nanoparticles for larval toxicity test.

Nanoparticle	24 h LC50 (mg/L)	48 h LC50 (mg/L)	
Ag	10.12		
AgNO3	8.190		
Au		7.54	
SiO2		23.59	
TiO2		18.94	
CuO		20.92	
CuSO4		13.45	
ZnO		259.34	
ZnSO4		0.796	

Figure 4 Mortality of nauplii in standard ARC test.

Silver nanoparticles and silver in ionic form (A), show the highest toxicity in a concentration-dependent manner. All the nanoparticles show concentration-dependent mortality, gold NP (B), CuO (C), ZnO (D), SiO2 (E) and TiO2 (F).

In the ARC test; we do see sedimentation and flocculation of nanoparticles by 48 h. The nauplii maintained in SW containing nanoparticles showed accumulation of NPs in the gut of Artemia by end of 24 h (Fig. 5). In case of silver nanoparticles, the nauplii that were dead within 24 h, the gut does not show accumulation of material (Fig. 5), even the nauplii appeared stunted. Staining of the nauplii exposed to nanoparticles showed signs of stress when stained with DCFDA for ROS and DHE for Superoxide (Fig. 5). Nauplii exposed to silver nanoparticles showed highest staining, followed by TiO2 and SiO2. Nauplii exposed to other nanoparticles do not show any signs of stress did show accumulation of nanoparticles in the gut (Fig. 5).

Figure 5 ROS staining of dead nauplii.

Nauplii of Artemia in standard ARC test stained with DCFDA (green) for ROS and DHE (Red) for superoxide accumulation due to stress. Nauplii exposed to silver NPs, TiO2 and SiO2 NP show higher staining for both ROS and superoxide at 24 h. (A–D) Control; (E–H) silver NP; (I–L) gold NP; (M–P) CuO NP; (Q–T) ZnO—NP; (U–X) titania and (Y–BB) SiO2NP.

Discussion

In this study, we have compared hatching rate of Artemia cyst in presence of nanoparticles and the toxicity of nauplii using standard Artemia test at various concentrations of nanoparticle suspension in high salinity conditions. We found a strong correlation in hatching rates and larval mortality in various nanoparticle suspensions (Fig. 6).

Figure 6 Correlation graph.

Regression analyses to show the correlation between two sets of results. There is a strong inverse correlation between the hatching rate and the mortality in the standard Artemia test (coefficient of correlation −0.748).

Artemia are able to tolerate high salinity and adapt to extreme conditions. They have wide distribution globally and the collection of cysts is easy. Artemia cysts are dormant gastrula states during development encased in hard outer shell; formed under unfavorable conditions. Under simple laboratory conditions, the cysts can be hatched in high salinity water when by about 20 h the umbrella stage emerges and yields free swimming nauplii (Sorgloos, Van Der Wielen & Persoone, 1978; Ward-Booth & Reiss, 1988). In the hatching test, we counted the number of umbrella stage and nauplii after 24 h, among the hatched larvae, number of motile nauplii were recorded for each tested nanoparticle. Hatching or emergence of the nauplii has been reported earlier as a sensitive bioassay to study the molecular and physiological effects of metals (Migliore et al., 1997; MacRae & Pandey, 1991). Lower hatching rate equates with the toxicity of test compound. Hatching has been used as an end point by some more laboratories while studying toxicity (Migliore et al., 1997; Rotini et al., 2015; Sarabia et al., 2008). Hatching of cysts can vary depending on a number of factors like geographical origin of cysts, salinity of hatching media, temperature, luminosity, pH etc. (Libralato et al., 2016) Despite all the factors, reduction in emerging nauplii during hatching has proven to be a reliable end point for toxicity studies.

Keeping the salinity at 2.5%, pH above 8 and temperature constant in all experimental conditions, we show that hatching rates are affected in presence of nanoparticles of metals and metal oxides. Silver and copper oxide nanoparticles show high toxicity among the various particles tested in concentration-dependent manner. To rule out the toxicity due to released ions from the nanoparticles, the nanoparticle suspension was centrifuged and resuspended in brine solution (Jemec et al., 2016). These resuspended particles were seen as nanoparticles even after 72 h in high salinity water and cause toxicity by affecting both hatching rates and viability. Though ionic silver precipitated out due to salt in water, it continued to affect the hatching and subsequent viability of the nauplii. Solutions of silver nitrate, copper sulfate and zinc sulfate representing ionic forms also showed high toxicity. Hatching of cysts can be affected by lower pH (Libralato et al., 2016) but our studies showed that the pH of SW containing nanoparticles also remained above 8. Other nanoparticles like; gold, TiO2 and SiO2 nanoparticles had some effect on hatching rates and survival after hatching in concentration-dependent manner. SiO2 nanoparticles at low concentration were very conducive to and in fact showed better hatching as compared to control (Table S1). Hatching rates in presence of nanoparticles were not much different at 20 and 24 h but the mortality of hatched nauplii was higher in some nanoparticles at 24 h (data not shown). Locomotion of the hatched nauplii was also affected in presence of nanoparticles.

In the standard Artemia test, nauplii hatched in SW are exposed to nanoparticle suspensions. The mortality and toxic effects expressed as LC50; the concentration of an agent at which 50% of the tested animals are dead after 24 h; were chosen as criteria of the toxicity (Nunes et al., 2006). The standard Artemia toxicity test results were comparable and silver nanoparticles ranked very high in toxicity, followed by copper and zinc oxide nanoparticles. TiO2, gold and SiO2 nanoparticles in that order, showed a better survival of nauplii up to 48 h. On comparing the two methods, a good inverse correlation was observed between the results obtained, that is, effects of NPs on hatching rates and effect on mortality of the hatched larvae (r = −0.79). The particles that reduced the hatching rate showed an increase in mortality in the ARC test using the normally hatched nauplii. Our results correlate well with some previous reports, where toxicity assays were performed on Artemia using nanoparticles (Lee, Chen & Chou, 1999; Rajasree et al., 2011; Ates et al., 2013). Comparing the viability of nauplii (inverse of mortality) in ARC test with the hatching rate showed no statistically significant difference in the mean values for each concentration of different particles (p = 0.683) in the paired t-test.

The advantage of the hatching assay is that it is easy to perform, short term and uses mortality as the end point for assessing toxicity of the test material. The question of altered solubility may not arise though sedimentation of these suspensions sometimes may vary for different particles. So far, we have seen that nanoparticles tested in this work remain stable in seawater up to 1 week, though the ionic or dissolved component may settle down due to precipitation. The sedimentation factor is taken care of by the continuous aeration in the hatching test that keeps the particles in suspended form.

Organisms in aquatic environment like daphnia incorporate NPs via gut. Artemia is also a filter feeder and the NPs may enter the guts of Artemia through ingestion. The mortality observed may be due to the uptake of the NPs clogging the gut as most of the nauplii in presence of NPs show presence of particles in the digestive tract. In the hatching test, the dead Artemia and partially hatched cysts show increased oxidative stress in presence of nanoparticles as indicated by DCDFA staining. Even the nauplii exposed to some nanoparticles like silver NPs showed accumulation of ROS and superoxide but with remaining nanoparticles it is likely that the gut accumulation of the NPs is the cause of mortality.

Artemia with capacity to homeostasis in variable external salt concentrations certainly appears to be a model organism to assess the nanoparticle behavior and ecotoxicity. Suitability of Artemia for toxicity tests is established and is often used in pharmaceutical industry and for metal and radiation toxicity (Pelka et al., 2000; Dvorak & Benova, 2002; Mayorga et al., 2010). We propose that Artemia hatchability test along with standard lethality/toxicity test definitely could also be used as prescreening of nanoparticle toxicity prior to their validation through in vivo toxicity studies. These acute toxicity tests eliminate the need of laboratory maintenance of test species which otherwise take up space and time. Use of cysts which are available in enormous quantities as batch also cut down on the variability in test species.

Conclusions

Hatchability of Artemia can be used as a quick screening test for toxicity of materials without much need for laboratory space and maintenance. With strictly laid out protocols and test conditions, this assay can be adopted for the hazard identification of nanomaterials and high throughput methods.

Supplemental Information

Supplemental Information 1 Supplementary data for toxicity of Nanoparticles.

Figure S1 The pattern of distribution of particles in MilliQ and salt water for three days. DLS measurements were made up to 48 h. We show an increase in aggregates size for some of the nanoparticles. Except for TiO2 and SiO2; most of the particles remain suspended and do not show visible sediment.

Figure S2 Images of Artemia hatched in presence of nanoparticles a. control, b, g-silver nanoparticles, c-CuO nanoparticles. Nauplii do not show any gross abnormality but often one sees still-dead nauplii enclosed in the membrane (e, f) where appendages are not free.

Figure S3 Artemia hatched in the presence of nanoparticles, were stained for reactive oxygen species using DCFDA, and Dead nauplii showed the staining in the gut (V), and the whole body in still–dead unhatched nauplii with membrane [C–F].

Supplementary table Data of individual experiments for hatching of Artemia in presence of SiO2NP. At 0.011 mg/L concentration the percent hatching of Artemia was always more as compared to hatching in salt water. (data not presented in the paper)

Click here for additional data file.

Supplemental Information 2 Raw data for toxicity of Artemia sp.

Toxicity of Artemia was assessed by hatching cysts in salt water containing metal and metal oxide nanoparticles and percent hatching was used as an endpoint (sheet 2 & 3–Fig. 3). The results of toxicity in hatching assay was compared with toxicity of Artemia in standard Artemia test using nauplii (Fig. 4). Size and distribution of particles in salt water and milliQ water is shown in (sheet 1 and sheet 5–Fig. 2).

Click here for additional data file.

Additional Information and Declarations

Competing Interests

Author Contributions

Data Availability

The authors declare that they have no competing interests.

Rohit Rekulapally performed the experiments.

Lakshmi Narsimha Murthy Chavali performed the experiments.

Mohammed M. Idris analyzed the data, contributed reagents/materials/analysis tools, prepared figures and/or tables, approved the final draft.

Shashi Singh conceived and designed the experiments, analyzed the data, contributed reagents/materials/analysis tools, prepared figures and/or tables, authored or reviewed drafts of the paper, approved the final draft.

The following information was supplied regarding data availability:

The raw data is available in the Supplemental Files.

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
