# Peer review of "Toxicity of TiO2, SiO2, ZnO, CuO, Au and Ag engineered nanoparticles on hatching and early nauplii of Artemia sp"

_PeerJ, doi:10.7717/peerj.6138_

## Round 0.1 · original submission · Major Revisions

As you can see, the Reviewers have provided substantial comments on your manuscript. Please consider revising your manuscript according to the suggestions, so that the manuscript can be further considered for publication.

Also, please excuse some of the language in the Reviewer reports, I believe it is due to the Reviewers being non-native English speakers.

After you have carefully addressed all the Reviewers’ comments and concerns, and edited the manuscript accordingly, the manuscript will likely be sent for a second round of peer review.

Reviewer 1 ·

Basic reporting

The quality of the English needs to be improved. More relevant literature review should be added in the Introduction and Discussion to give a clear rationale of this study and explain the results better. All the figures need polishing. Remove the grid lines in the bar charts and increase the resolution.

Line 35 The abstract should summarize and refine the most significant findings in the study, rather than list the experiments and assays that have been done.

Line 50 The author should explain the rationale that why they chose these six metal and metal oxides nanoparticles in this study. The relevant literatures should be reviewed in the introduction.

Experimental design

No information about the DHE assay displayed in Figure 5 was mentioned in the Method section.

Line 86 “4 metal oxide nanoparticles (TiO2, SiO2 and CuO)”, should include “ZnO”.

Line 129 “Five types of nanoparticles (Ag, Au, SiO2, TiO2, CuO and ZnO) are tested”, should be “Six”.

Validity of the findings

The data is reasonable, but do not provide much of insights.

Line 172 & 174 The description of the size as “higher” is very ambiguous, the author should report the exact size of each particle in the text.

Line 203 “hatching rate was better…” crappy English, hatching rate was higher…

Line 240 More explanation should be added in the Discussion to reveal more insight about the data.

Line 258-259, Line 266 What is the mechanism of the hatching interference observed in the study? The author claimed that the metal ion release could be a reason for silver and copper oxide nanoparticles, so they centrifuged the nanoparticles, resuspended in the solution and considered them as “nanoparticles” after 72 h. The authors should provide evidence for this statement first, if this statement is correct, what could be the reason for the hatching interference in these non-dissolvable nanoparticles? The authors also reported that “Silicon nanoparticles at low concentration were very conducive in fact showed better hatching as compared to control.” More clear explanation should be added in this part.

Additional comments

This paper describes the toxicity of six different metal and metal oxides nanoparticles TiO2, SiO2, ZnO, CuO, Au, and Ag on the hatching and early nauplii of Artemia sp.. However, the paper is very descriptive and superficial and it fails to explain the data in terms of the physiology of the organism in any detail. The quality of the English, although reasonable, was quite clumsy in places. The figures need a lot of polishing for a publication. Overall, the manuscript won’t be suggested to be published in the journal in current form.

Reviewer 2 ·

Basic reporting

The manuscript addresses an important subject. The authors examined the effect of 4 metal oxide and 2 metal nanoparticles on hatching and early nauplii of Artemia sp. in order to demonstrate the reability of the hatching assay.
The interest and justification of the paper is not very accurate and I have some comments in order to improve the manuscript

Experimental design

I have some concerns about the experimental design. This section needs a lot of refinement. The experimental set-up is poorly written in that it is very unclear.

Validity of the findings

There is not an adequate statistical analysis.
There is a weakness in the experimental design, it results confuse and not detailed

Additional comments

Introduction
The Introduction could use a bit more of a background on work that has been done in particular evidencing the recent work on this focus.
These papers can be an example:

-Libralato, G., Prato, E., Migliore, L., Cicero, A.M., Manfra, L., 2016. A review of toxicity testing protocols and endpoints with Artemia spp. Ecol. Indic. 69, 35–49.
-Libralato, G., 2014. The case of Artemia spp. In nanoecotoxicology. Mar. Environ. Res. 102, 38–43.
-A. Rotini, A. Gallo, I., Parlapiano, M.T. Berducci, R. Boni, E., Tosti, E. Prato, C. Maggi, A.M. Cicero,
L. Migliore, L. Manfra, 2018. Insights into the CuO nanoparticle ecotoxicity with suitable marine model species Ecotoxicology and Environmental Safety 147: 852–860.

-A. Rotini, L. Manfra, S. Canepa, A. Tornambè, L. Migliore, 2015. Can Artemia Hatching Assay Be a (Sensitive) Alternative Tool to Acute Toxicity Test? Bulletin of Environmental Contamination and Toxicology. 5(6) 745-751.

Materials and Methods
I have some concerns about the experimental design. This section needs a lot of refinement. The experimental set-up is poorly written in that it is very unclear.  

Line 101 – Please use chemical symbols adequate.

line 103 –Precise which means PVP, since it is not referred previously.

Line 104-“Working dilutions of the nanoparticles are made in salt water (NaCl) for various studies” re-write this sentence is not clear.

-The authors measured the sizes nanoparticle suspension at 25μg/ml in milliQ water and salt water, but for toxicity test they used different concentrations of each nanomaterial (from 100mg/L to 0.01mg/L Why?

-I think it is needed at least more time points of for the NP dissolution (if possible a graph with the time evolution of the dissolution during the 3 days) and DLS of the NPs at different time points in the working media (with characteristics similar to the one used. Having a better characterization of the evolution of the NPs in the working media can be valuable for the scientific community.

-Please specify the preparation of stock suspension. In M&M is not reported about of three soluble salts : CuSO4, ZnSO4 and AgNO3.
-Line 145 LD50 change with LC50

-There is not an adequate statistical analysis.

Results
This section is not very clear the data are not well explained.

-The analysis of nanoparticles need a statistical analysis (one way ANOVA) to determine statistical significance among the treatments (distilled and sea water).

-Figure 1 are not ordered according to the text.

-Line 179 “The nanoparticles suspended both in Milli Q water and 25%NaCl solution (salt water) were measured up to 48 hours to see change in their dispersion pattern”. But the nanoparticle suspensions (in milliQ water and salt water) were measured thrice for three day???
-Line 187 “Brine shrimp normally showed  a hatching rate of 74%+5.5, the viability of these hatched nauplii was about 97%.” Pleas give a reference
-Line 191-193 “The hatching rate dropped to about 29% in the two nanoparticles; of these only 50% were motile and viable at a concentration of 100mg/L (Table 1)”.
Please give a graph or table with the viability of hatched nauplii.
-Why the authors reported also values of hatching rate for silver nitrate, copper sulfate, zinc sulfate. Maybe they use them reference toxicants? Give more information about this.

-Line 199-200 I don’t agree with this sentence, moreover the fig 4 referred on mortality data. Please, to compare the data use statistical analysis.
-Line 208 “The pH of salt water did not change upon …..” Please in M&M section reports how the authors measure pH , t°, salinity, oxygen.
-Line 211- Where is the Figures S2.1-2.6???

-Line 213 – Where is the Fig. S3. In the presentation of tables and figures I belive that there is a lot confusion.
-Line 221- “The most toxic was the silver nanoparticles and silver in ionic form”
Table 2 reported the lethal concentration only of nanoparticles for larval toxicity test. I do not see the silver in ionic form!!! The relased ions is crucial in determining the toxicity of NP suspension and for this purpose the ion concentration must be determined and the metal release was examined at 0, 24 h, 48 h, 72 h, and 96 h , after the stock suspensions were prepared.
-Line 225-“LC50 of gold was 2.53E-226  03mg/Lat 48h”. There is some prolem.

-Line 226. Order the text according to the increase value or according to the table.
Discussion
-Line 243. 25% is a high salinity condition??? I do no belive
-Line 243. This aspect is not reported in the results.
-Line 257. Together at silver and copper oxide also ZnONP showed high toxicity.
- Line 258- 259 As reported above, this aspect must be added to the M&M
- Table 1 is not acceptable in this form: lack the units, please indicate the gold with capital letter (as Au), moreover added a column where to put the used concentrations.
-Table 2. In the column change with 24h LC50 and 48h LC50 because it is not a percent, the LC50 is a concentration of a chemical that determines the death of 50 percent of the sample population.

-Figure 1 Make more readable

-Figure 2 Please, use the chemical symbols (also in fig 3)

-Figure 3 It is very bad, to lack of units

-Figure 4 Report the same graduation for all graphs

-resolution of Figure 5 is too low.
The English language should be improved

---

## Round 0.2 · Minor Revisions

While the manuscript has been improved and is scientifically sound, before I can make the decision, I am asking you to proof read the manuscript or have a native English speaker proof read the text. Please pay special attention to capitalization of words (for example, “nauplii” seems to be used both with and without capitalization), letters and numbers that need to be either in subscript or superscript (for example, in chemical formulas) and consistent use of acronyms (the use of eNanoparticles or eNPs is confusing, please harmonize to NPs throughout the text and figures/tables). Also, the number of decimal places reported for values should be critically reviewed. For example, in lines 188-192 hydrodynamic diameters of different nanoparticles are reported with either two, one or zero decimal places. Please decide how many digits to the right of the decimal point are meaningful and harmonize. Considering the variability of the DLS method, and the size range of the particle agglomerates (in micrometers), the digits to the right of the decimal point are probably not meaningful. Same comment on decimal places goes also to other values in the text as well as tables. Please make sure that the Figure panels are all in the same size, use the same font size and are aligned. Check the correctness of axis labels (including capitalization of words and the use of units, especially in Fig 2 and 4). Explain in the figure captions the use of acronyms or NP concentrations if it is not evident from figure legends or axis labels, like in Fig 3 and 4. In the figure caption, explain what is shown in each panel, instead of commenting on the results which should be explained in the text. Please check that Table titles are above the tables. After correcting the language, figures and tables the manuscript will be acceptable for publication.

Reviewer 1 ·

Basic reporting

The English still needs more polishing.

Experimental design

no comment

Validity of the findings

no comment

Additional comments

All the comments in the first round have been addressed. However, there are still some problems need to be fixed in a scientific article.
Figures 2 and 3 need to be improved, they look fuzzy and ugly;
The citation format in the text needs to be consistent. Line 290, Line 300;
Line 197, the abbreviation of milliQ water needs to be consistent;
All the supplemental items need to put in a word file, with detailed figure legends.

Reviewer 2 ·

Basic reporting

No comment

Experimental design

No comment

Validity of the findings

No comment

Additional comments

I have read the review prepared by the authors.
The work has certainly been improved in some aspects.

---

## Round 0.3 · accepted · Accept

Thank you for carefully revising the manuscript. I look forward to reading the published version of your article and wish you all the best in your future research.

#